

# Reliability of hip muscle strength measured in principal and intermediate planes of movement

Basilio A.M. Goncalves[1,2], David J. Saxby[1,2], Adam Kositsky[1,2], Rod S. Barrett[1,2] and Laura E. Diamond[1,2]

[1] School of Health Sciences and Social Work, Griffith University, Gold Coast, Queensland, Australia
[2] Griffith Centre of Biomedical and Rehabilitation Engineering (GCORE), Menzies Health Institute Queensland, Griffith University, Gold Coast, Queensland, Australia

## ABSTRACT

**Background.** Muscle strength testing is widely used in clinical and athletic populations. Commercially available dynamometers are designed to assess strength in three principal planes (sagittal, transverse, frontal). However, the anatomy of the hip suggests muscles may only be recruited submaximally during tasks performed in these principal planes.
**Objective.** To evaluate the inter-session reliability of maximal isometric hip strength in the principal planes and three intermediate planes.
**Methods.** Twenty participants (26.1 ± 2.7 years, 50% female) attended two testing sessions 6.2 ± 1.8 days apart. Participants completed 3-5 maximal voluntary isometric contractions for hip abduction, adduction, flexion, extension, and internal and external rotation measured using a fixed uniaxial load cell (custom rig) and commercial dynamometer (Biodex). Three intermediate hip actions were also tested using the custom rig: extension with abduction, extension with external rotation, and extension with both abduction and external rotation.
**Results.** Moderate-to-excellent intraclass correlation coefficients were observed for all principal and intermediate muscle actions using the custom rig (0.72–0.95) and the Biodex (0.85–0.95). The minimum detectable change was also similar between devices (custom rig = 11–31%; Biodex = 9–20%). Bland-Altman analysis revealed poor agreement between devices (range between upper and lower limits of agreement = 77–131%).
**Conclusions.** Although the custom rig and Biodex showed similar reliability, both devices may lack the sensitivity to detect small changes in hip strength commonly observed following intervention.

Corresponding author
Basilio A.M. Goncalves,
b.goncalves@griffith.edu.au

## INTRODUCTION

Deficits in hip muscle strength are common in a broad range of musculoskeletal conditions, including femoroacetabular impingement (FAI) syndrome (*Casartelli et al., 2011*; *Diamond et al., 2015*), hip osteoarthritis (*Arokoski et al., 2002*), adductor-related groin pain (*Hölmich et al., 1999*), knee ligament injuries (*Khayambashi et al., 2016*), chronic ankle instability (*McCann et al., 2018*), and low back pain (*De Sousa et al., 2019*). Hip muscle strength is

also considered a determinant for athletic performance, as elite players show larger hip strength compared to sub-elite players (*Prendergast et al., 2016*). Muscle strength is typically assessed using commercially available dynamometers during maximal voluntary muscle contractions and compared with earlier testing sessions, the unaffected limb, and/or a control group (*Kierkegaard et al., 2018*). Although a motor-driven dynamometer (MDD) is considered current best practice for measurement of muscle strength (*Desmyttere, Gaudet & Begon, 2019*; *Martins et al., 2017*; *Thorborg, Bandholm & Hölmich, 2013*), these devices are large, expensive, and limited to measurements in three principal planes (i.e., sagittal, frontal, and transverse).

Hand-held dynamometers (HHD) are a portable and reliable alternative to an MDD, but measurements are highly dependent on the skill and relative strength of the assessor (*Thorborg et al., 2011*). Externally-fixed dynamometers minimise influence of the assessor (*Desmyttere, Gaudet & Begon, 2019*; *Martins et al., 2017*; *Thorborg, Bandholm & Hölmich, 2013*) and have potential to measure strength in any movement plane, but are mainly used to measure force in principal planes. Given the oblique lines of action the posterior/lateral hip muscles (e.g., piriformis, gluteus medius) (*Neumann, 2010*), maximal contraction in planes outside the principal planes may be necessary to generate the highest levels of activation, as previously shown at the knee joint (*Buchanan & Lloyd, 1997*). Thus, testing maximal strength in the principal planes alone may not elicit maximal muscle activation, and consequently may be unable to fully evaluate the function of the posterior/lateral hip musculature.

Increasing hip strength may effectively restore function in individuals with hip osteoarthritis (*Bennell & Hinman, 2011*) and adductor-related groin pain (*Hölmich et al., 1999*), however, strength training appears less effective for individuals with FAI syndrome (*Casartelli et al., 2018*). This discrepancy in therapeutic effectiveness could relate not only to the complex anatomical structures comprising the hip joint and requirement for control over six degrees of freedom (*Neumann, 2010*), but also to the limited understanding of deep hip muscle function (*Diamond et al., 2016*). Pain and reduced range of motion in movements combining hip flexion, adduction, and internal rotation are commonly reported in FAI syndrome (*Diamond et al., 2015*). Hip extensor, abductor, and external rotator muscles have potential to directly oppose motions of impingement (*Neumann, 2010*) and to influence hip joint loading during locomotive tasks (*Catelli et al., 2019*). As the hip moves into extension, the potential of muscles like piriformis, gluteus medius, and gluteus maximus to produce external rotation torque increases (*Delp et al., 1999*). Hip rotation angle may also affect the force generation capacity of the larger gluteal muscles (*Delp et al., 1999*; *Ward et al., 2009*; *Ward, Winters & Blemker, 2010*). Thus, assessment of muscle force measurements in intermediate planes combining hip extension, abduction, and external rotation may better quantify hip muscle strength in those with FAI syndrome.

Given that MDDs are limited to measurements of maximal hip strength in the principal planes, we constructed a cost-effective custom rig consisting of a metal frame and load cell in order to measure maximal hip strength in both the principal and intermediate (i.e., oblique to the principal planes) planes with minimal set-up requirements. The primary aim of this study was to evaluate the inter-session reliability of isometric hip

force and torque measurements acquired using the custom rig. The secondary aim was to compare hip flexion, external rotation, and internal rotation torque values and reliability indexes obtained from the custom rig with measurements from an MDD. We hypothesized that hip muscle strength could be reliably assessed in both the principle and intermediate planes using the custom rig, and that the custom rig would have similar measurement error to an MDD.

## MATERIALS & METHODS

### Participants

Twenty recreationally active adults (age $= 26.0 \pm 2.5$ years, range $= 23–33$ years, female $= 50\%$, mass $= 69.9 \pm 14.0$ kg, height $= 1.7 \pm 0.1$ m, body mass index $= 23.1 \pm 3.0$ kg/m$^2$) with no history of hip surgery in their lifetime nor lower limb injury in the past three months were recruited from the university community to participate in this study. Our sample size was determined based on recommendations from *Walter, Eliasziw & Donner (1998)*, with requirements for a good level of reliability (an interclass correlation coefficient (ICC) >0.75 (*Koo & Li, 2016*)) and the acquisition of three trials per participant. The study was approved by Griffith University's human research ethics committee (GU Ref No: 2018/700) and all participants provided written informed consent. The study adhered to the Guidelines for Reporting Reliability and Agreement Studies (GRRAS) (*Kottner et al., 2011*).

### Procedures

Participants attended two testing sessions separated by 3–9 days ($6.2 \pm 1.8$ days) and were asked to refrain from lower body resistance or unfamiliar exercise in the 24-hours preceding each testing session. Following a 5-minute self-paced warm-up on a cycle ergometer with a 10N resistance, participants completed a sequence of maximal voluntary isometric contractions (MVIC) on (i) a custom rig - consisting of a uniaxial load-cell (Dacell UU-K500, Korea, capacity $= 4903$N $\pm1\%$, cost $= \sim400$USD) with cable and strap fixated to a metal frame that was rigidly coupled to the floor (Fig. 1) and (ii) an MDD (System 4 Pro, Biodex Medical Systems, NY, USA) (Fig. 2). All participants performed MVICs using their dominant leg (i.e., leg used to kick a ball). Participants rested for 10 min when moving between devices to minimise neuromuscular fatigue. Testing device and hip action orders were randomised between participants. The same device and contraction orders were used in both testing sessions. Participants completed two submaximal familiarisation contractions (at their own discretion) followed by a minimum of three MVICs for each hip action with standardised verbal encouragement. Further attempts were performed when measured force increased by >5% in the third trial compared to the first and/or second trial (*Peltonen et al., 2018*). Force did not increase by >5% after the second attempt in >65% of participants and no participant performed more than five attempts. A minimum of 30 s and two minutes rest was allowed between contractions and between different muscle actions (e.g., abduction or adduction), respectively (*Thorborg, Bandholm & Hölmich, 2013*).

Participants performed six maximal isometric hip strength tasks in the principal planes of movement using the custom rig and MDD (Figs. 2A–2D). Additionally, three intermediate

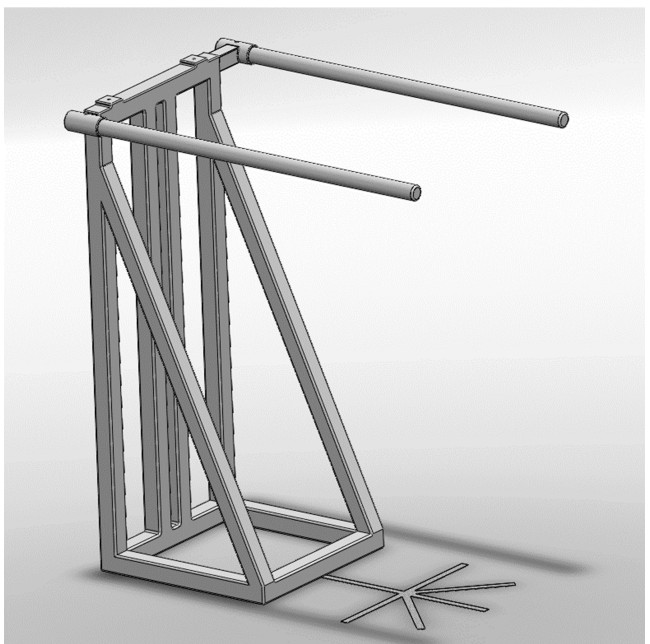

**Figure 1** Schematic representation of the metal frame used to fixate the load cell.

muscle actions were tested using the custom rig only: (a) combined hip extension and abduction (E+AB); (b) combined hip extension and external rotation (E+ER); and (c) combined hip extension, abduction, and external rotation (E+AB+ER) (Fig. 2E). Strength testing body positions were chosen with consideration of the physical constraints of each device. Hip abduction and adduction strength were tested in a supine position using the custom rig and in a side-lying position using the MDD. For both devices, knees were placed in neutral (0° of flexion) and the hips were positioned in 15° of abduction and neutral rotation. Hip flexion, extension, and internal and external rotation strength were tested in the same position for both devices. Hip flexion strength was tested in a supine position with the hip and knee in 45° and 90° of flexion, respectively. Hip extension was tested in a prone position with the hip and knee in neutral (0° of flexion). Hip internal and external rotation were tested in a sitting position with the hip and knee both in 90° of flexion. The three intermediate muscle actions (E+AB, E+ER, and E+AB+ER) were assessed in a standing position with the hip and knees in neutral (0° of flexion). For E+ER and E+AB+ER, participants placed their hips at their maximum range of hip external rotation. For E+AB and E+AB+ER, participants were instructed to produce force at a 45° angle (midway between pure abduction and pure extension), and for E+ER participants were instructed to produce force in the direction of pure extension (Fig. 2E). The strap was attached to the distal shank (as close as possible to the lateral malleoli) for hip abduction, adduction, extension, internal and external rotation, and all intermediate plane measurements using the custom rig, and for hip internal and external rotation measurements using the MDD. For all other positions, the strap was attached to the distal thigh (as close as possible to the lateral femoral condyle). For measurements acquired using the custom rig, joint angles

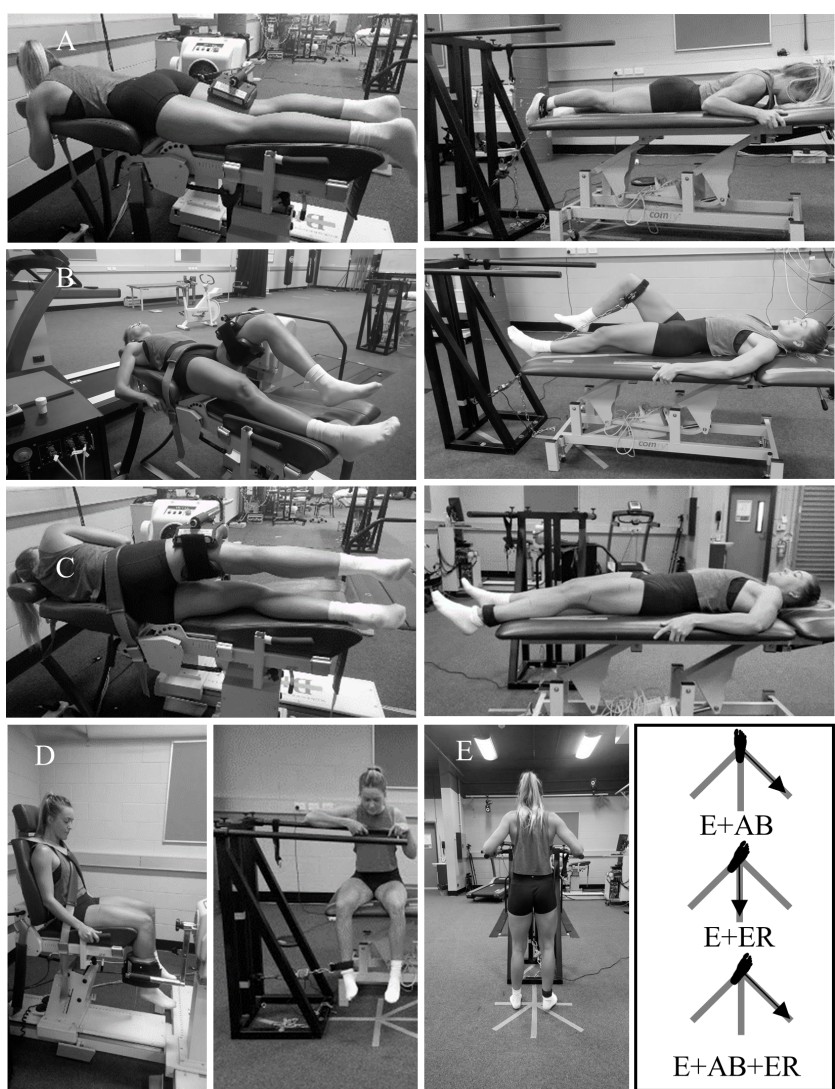

**Figure 2** **Hip muscle strength testing positions using a motor-driven dynamometer (MDD) and a custom rig.** (A) hip extension using the MDD (left) and custom rig (right); (B) hip flexion using the MDD (left) and the custom rig (right); (C) hip abduction/adduction using the MDD (left) and the custom rig (right); (D) hip internal/external rotation using the MDD (left) and the custom rig (right).

were confirmed manually with a goniometer (*Buchanan & Lloyd, 1997*). For hip extension using the custom rig, the angle between the direction of force production (vertical) and the direction of force measurement (30° with respect to vertical) (Fig. 2A, right) was controlled for all participants and used to calculate the vertical component of force (hip extension force = measured force/cos (30°)). For measurements acquired using the MDD, joint angles were confirmed with the device's inbuilt goniometer and the greater trochanter was aligned with the device's axis of rotation. The inbuilt goniometer was calibrated with a digital inclinometer before each task to offset the extra padding in the MDD. For hip abduction and adduction, the axis of rotation of the MDD was aligned with the intersection
of a vertical line crossing the anterior superior iliac spine and a horizontal line crossing the greater trochanter (*Sugimoto et al., 2014*). For hip internal and external rotation, the axis of rotation was aligned with the centre of the patella (Fig. 2D). The weight of the test limb was recorded and used to correct for gravity (*Buchanan & Lloyd, 1997*). Measurements acquired in the principal planes using the custom rig were converted to torque (N.m) by multiplying the measured force (N) by the lever arm (m). The lever arm was measured from the most lateral bony prominence of the greater trochanter to the centre of the attachment point (i.e., strap) for each position. For internal and external rotation, the lever arm was measured from the centre of the patella to the centre of the attachment point.

## Data processing

All data were recorded with Vicon Nexus 2.7.1 software (Vicon, Oxford Metrics Group, UK) at 1,000 Hz. Data were subsequently low-pass filtered at 6 Hz using a dual pass 2nd order Butterworth filter. The highest force or torque value from each muscle action was used to determine maximum strength.

## Data analysis

We excluded statistical outliers from the analysis following established procedures (*Kwak & Kim, 2017*) and confirmed data normality using the Shapiro–Wilk test (*Ghasemi & Zahediasl, 2012*). We determined inter-session reliability (*Atkinson & Nevill, 1998*) using ICC with a 95% confidence interval (CI), based on a single measurement, absolute-agreement, two-way mixed-effects model (*Koo & Li, 2016*). ICCs <0.5 were interpreted as poor, $0.5-0.74$ as moderate, $0.75-0.89$ as good, and $\geq 0.9$ as excellent (*Koo & Li, 2016*). We also estimated absolute reliability (*Atkinson & Nevill, 1998*) (or measurement error) using the standard error of measurement (SEM) (the standard deviation of the between-day difference in strength/$\sqrt{2}$) (*Weir, 2005*) and minimal detectable change (MDC) based on a 90% CI ($\text{MDC}_{90}$ = standard deviation of between-day difference in strength $\times 1.65$) (*Weir, 2005*). Both SEM and MDC were calculated as a percentage (SEM% and MDC%) by dividing each absolute value by the grand mean for that strength variable. Reliability and measurement error were calculated for torque and force measurements to ensure the lever arm measurements had no influence on the strength assessment. We assessed the agreement between hip flexion, internal rotation, and external rotation strength using the custom rig and MDD using the Bland-Altman method with data from both testing sessions ($n = 40$) (*Bland & Altman, 2010*) by plotting the percentage change in torque between devices ((custom rig-MDD)/MDD$\times 100$) against the mean of the two strength measures. Bias and limits of agreement (LoA) were calculated as the mean difference in torque between devices and standard deviation of the mean differences multiplied by 1.96, respectively. The CI for the limits of agreement were calculated as previously reported (*Carkeet & Goh, 2018*). We did not test agreement between devices during hip abduction, adduction, or extension given the differences in body position and lever arm between devices, which have been shown to affect absolute force values (*Thorborg et al., 2010*). All analyses were undertaken using custom scripts in MATLAB R2018a (The MathWorks, Inc., Massachusetts, USA).

**Table 1    Reliability metrics for hip torque measurements using a custom rig.**

|  | Session 1 (N.m) | Session 2 (N.m) | ICC (95% CI) | SEM (95% CI) (N.m) | SEM % | MDC (95% CI) (N.m) | MDC % |
|---|---|---|---|---|---|---|---|
| Abduction ($n = 20$) | $95 \pm 21$ | $99 \pm 24$ | 0.77 (0.51–0.90) | 11 (8–16) | 11 | 25 (19–37) | 26 |
| Adduction ($n = 20$) | $124 \pm 29$ | $126 \pm 26$ | 0.92 (0.79–0.97) | 8 (6–12) | 6 | 19 (14–28) | 15 |
| Extension ($n = 18$) | $226 \pm 54$ | $210 \pm 52$ | 0.85 (0.57–0.95) | 19 (14–29) | 9 | 44 (32–68) | 20 |
| Flexion ($n = 17$) | $87 \pm 17$ | $88 \pm 17$ | 0.95 (0.84–0.98) | 4 (3–7) | 5 | 9 (7–15) | 11 |
| Internal rotation ($n = 20$) | $65 \pm 19$ | $65 \pm 17$ | 0.86 (0.66–0.94) | 7 (5–10) | 11 | 16 (12–24) | 25 |
| External rotation ($n = 20$) | $48 \pm 11$ | $50 \pm 13$ | 0.72 (0.41–0.88) | 6 (5–10) | 13 | 15 (11–22) | 31 |

**Notes.**

Values are mean ± standard deviation unless otherwise indicated.

ICC, intraclass correlation coefficient; CI, confidence interval; SEM, standard error of measurement; MDC, minimal detectable change; N.m, newton.metre; N, newton; E, extension; AB, abduction; ER, external rotation.

**Table 2    Reliability metrics for hip torque measurements using a motor-driven dynamometer.**

|  | Session 1 (N.m) | Session 2 (N.m) | ICC (95% CI) | SEM (95% CI) (N.m) | SEM % | MDC (95% CI) (N.m) | MDC % |
|---|---|---|---|---|---|---|---|
| Abduction ($n = 20$) | $117 \pm 22$ | $113 \pm 25$ | 0.85 (0.67–0.94) | 9 (7–13) | 8 | 21 (16–31) | 18 |
| Adduction ($n = 20$) | $90 \pm 24$ | $95 \pm 27$ | 0.89 (0.72–0.95) | 8 (6–12) | 9 | 19 (14–28) | 20 |
| Extension ($n = 19$) | $152 \pm 27$ | $155 \pm 28$ | 0.95 (0.86–0.98) | 6 (5–10) | 4 | 14 (11–22) | 9 |
| Flexion ($n = 20$) | $97 \pm 22$ | $102 \pm 25$ | 0.95 (0.74–0.99) | 4 (3–7) | 4 | 10 (7–16) | 10 |
| Internal rotation ($n = 19$) | $68 \pm 20$ | $65 \pm 17$ | 0.92 (0.78–0.97) | 5 (4–8) | 8 | 12 (9–18) | 18 |
| External rotation ($n = 20$) | $50 \pm 13$ | $52 \pm 15$ | 0.90 (0.75–0.96) | 4 (3–7) | 9 | 10 (8–16) | 20 |

**Notes.**

Values are mean ± standard deviation unless otherwise indicated.

ICC, intraclass correlation coefficient; CI, confidence interval; SEM, standard error of measurement; MDC, minimal detectable change; N.m, newton.metre; N, newton; E, extension; AB, abduction; ER, external rotation.

# RESULTS

We observed excellent reliability for torque measurements of hip adduction and flexion (ICC = 0.92−0.95) and good reliability for all other torque measurements (ICC = 0.77−0.86) using the custom rig, with the exception of external rotation (moderate reliability; ICC = 0.72) (Table 1). We observed good-to-excellent reliability for all torque measurements using the MDD (ICC = 0.85−0.95) (Table 2). Similarly, we observed good-to-excellent reliability of force measurements in all intermediate planes using the custom rig (ICC = 0.77−0.90) (Table 3). The SEMs ranged from 5–13% for the custom rig and 4–9% for the MDD. The MDCs ranged from 11–31% for the custom rig and 9–20% for the MDD.

From the Bland-Altman analysis we observed, on average, a greater torque measured by the MDD for hip flexion (9%), a greater torque measured by the custom rig for hip external rotation (1%), and similar values for hip internal rotation. The differences between the upper and lower limits of agreement were 94% for hip flexion, 131% for hip internal rotation, and 77% for hip external rotation (Fig. 3).

**Table 3  Reliability metrics for hip force measurements using a custom rig.**

| | Session 1 (N) | Session 2 (N) | ICC (95% CI) | SEM (95% CI) (N) | SEM % | MDC (95% CI) (N) | MDC % |
|---|---|---|---|---|---|---|---|
| Abduction ($n=20$) | 120 ± 26 | 124 ± 29 | 0.77 (0.51–0.90) | 13 (10–19) | 11 | 31 (26–35) | 25 |
| Adduction ($n=20$) | 158 ± 33 | 159 ± 28 | 0.89 (0.71–0.96) | 10 (8–16) | 7 | 24 (23–26) | 15 |
| Extension ($n=18$) | 287 ± 69 | 270 ± 66 | 0.90 (0.73–0.97) | 21 (16–33) | 8 | 49 (33–66) | 18 |
| Flexion ($n=17$) | 234 ± 42 | 239 ± 51 | 0.80 (0.53–0.92) | 21 (16–32) | 9 | 49 (44–53) | 21 |
| Internal rotation ($n=20$) | 171 ± 43 | 169 ± 39 | 0.81 (0.57–0.92) | 18 (14–27) | 11 | 42 (40–43) | 25 |
| External rotation ($n=20$) | 125 ± 25 | 130 ± 29 | 0.77 (0.48–0.91) | 13 (10–19) | 10 | 30 (25–35) | 23 |
| E+AB ($n=20$) | 208 ± 45 | 207 ± 52 | 0.86 (0.66–0.95) | 19 (14–29) | 10 | 45 (40–50) | 23 |
| E+ER ($n=20$) | 211 ± 54 | 196 ± 54 | 0.86 (0.68–0.94) | 18 (13–27) | 9 | 42 (41–42) | 20 |
| E+AB+ER ($n=20$) | 201 ± 55 | 197 ± 54 | 0.88 (0.70–0.95) | 20 (15–29) | 10 | 47 (32–62) | 23 |

**Notes.**

Values are mean ± standard deviation unless otherwise indicated.

ICC, intraclass correlation coefficient; CI, confidence interval; SEM, standard error of measurement; MDC, minimal detectable change; N, newton; E, extension; AB, abduction; ER, external rotation.

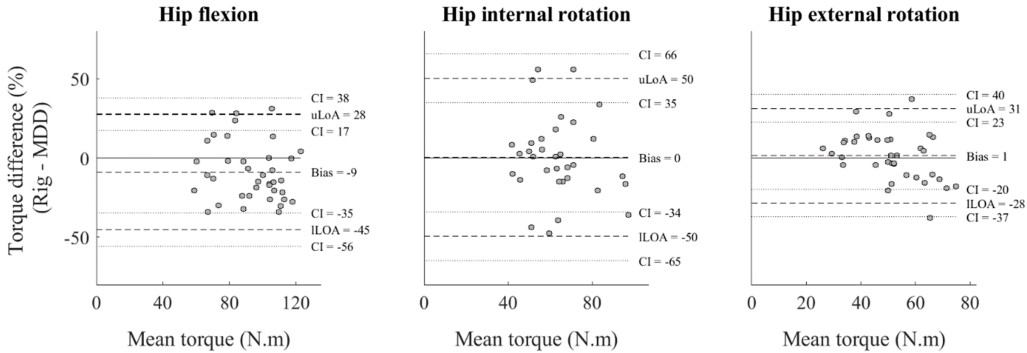

**Figure 3  Bland-Altman analysis of hip strength measurements.** Hip flexion (left), hip internal rotation (middle), and hip external rotation (right), showing differences in mean torque values between devices (custom rig)–motor-driven dynamometer (MDD); solid line), and upper (uLoA) and lower (lLoA) limits of agreement (dashed lines), with 95% confidence intervals (CI) (dotted lines)($n=40$).

## DISCUSSION

This study evaluated the inter-session reliability of a custom rig designed to measure hip muscle strength in both the principal and intermediate planes. We anticipated that if hip strength measurements obtained using the custom rig were shown to be reliable, the custom rig could contribute to improved diagnosis of, and treatment for, strength impairments in clinical and athletic populations. The custom rig showed good-to-excellent reliability for all muscle actions with the exception of hip external rotation, where we observed only moderate reliability. Similar reliability was observed with the MDD. However, we found large measurement error for most measurements on both devices, suggesting small changes in strength may not be detected. Further, the two methods showed poor agreement, suggesting that these devices should not be used interchangeably. Thus, although the custom rig may be used to measure maximal isometric force in principal and intermediate planes,

care must be taken when assessing hip external rotation or comparing values with those from a MDD.

Mean ICC values for torque measurements in the principal planes using the custom rig demonstrated excellent reliability for hip adduction and flexion good reliability for hip abduction, extension, and internal rotation, and moderate reliability for hip external rotation. These ICCs are similar to those observed for the MDD (0.85−0.95) and reported elsewhere for externally-fixed dynamometers (0.58−0.99) (*Bazett-Jones & Squier, 2020*; *Hickey et al., 2018*; *Katoh & Yamasaki, 2009*; *Martins et al., 2017*; *Romero-Franco, Jiménez-Reyes & Montaño Munuera, 2017*; *Thorborg, Bandholm & Hölmich, 2013*), hand-held (0.80−0.97) (*Thorborg et al., 2010*) or other user independent devices (0.76−0.99) (*Aramaki et al., 2016*; *Gonçalves et al., 2021*). The large CIs observed for our ICCs across strength measurements (95% CI [0.41–0.98]) may be a function of marginal between-participant variability (CV = 20–27%) (*Koo & Li, 2016*). Conceivably, tighter CIs could be achievable with a more heterogeneous sample. Although large, our CIs were comparable to previous reports from a range of measurement devices (95% CI [0.14–1.00]) (*Charlton et al., 2017*; *Thorborg, Bandholm & Hölmich, 2013*; *Thorborg et al., 2010*), suggesting the custom rig has similar reliability to other commonly used devices. Small differences in reliability between muscle actions are likely explained by the degree of familiarity of participants with these actions (*Hopkins, Schabort & Hawley, 2001*). Although our findings suggest the custom rig can reliably assess maximal strength in the principal planes in a healthy population, further research is needed to evaluate its use in clinical populations.

Our force measurements in the intermediate planes showed good reliability (ICC = 0.86−0.88 [95% CI [0.66–0.95]]). Compared to pure extension force (ICC = 0.90 [95% CI [0.73–0.97]]), the reliability of hip strength in the intermediate planes did not decline despite the novelty of the tasks. A single study investigated reliability of maximal isometric strength of hip abduction combined with hip external rotation ($n = 20$) using a belt-fixed HHD (*Aramaki et al., 2016*). The authors reported higher reliability (ICC = 0.97−0.98 [95% CI [0.94–0.99]]) than we observed for our measurements in the intermediate planes. However, in the previous study a clam testing position was used, which inherently assesses bilateral hip muscle strength, and only intra-day reliability was assessed (*Aramaki et al., 2016*). Furthermore, individuals with hip pathologies often present unilateral symptoms (*Agricola et al., 2014*), thus there is a need to measure unilateral hip muscle strength when conducting investigations in clinical populations. Although hip muscle strength performed in intermediate planes can be reliably measured, assessments of electromyography (*Glaviano & Bazett-Jones, 2020*; *Lee et al., 2014*; *McBeth et al., 2012*) are still needed to understand if these tasks do indeed recruit hip-spanning muscles maximally. In the future, the presented device could be used to explore the relationship between hip muscle strength measured in intermediate planes and the presence of hip symptoms and/or dysfunction to identify potential targets for rehabilitation.

The use of ICCs to assess reliability is influenced by between-participant variability (*De Vet et al., 2006*) and has been criticised across the literature (*Hopkins, Schabort & Hawley, 2001*; *Koo & Li, 2016*; *Thorborg, Bandholm & Hölmich, 2013*). Measurement error is more clinically relevant as it defines a cut-off value for meaningful change (*Thorborg,*

*Bandholm & Hölmich, 2013*; *Weir, 2005*). Measurement errors previously reported for hip strength measurements using hand-held (MDC ≈ 7–34%) (*Charlton et al., 2017*; *Thorborg et al., 2010*), externally-fixed (MDC ≈ 21–69%) (*Aramaki et al., 2016*; *Thorborg, Bandholm & Hölmich, 2013*), or other user-independent (MDC ≈14–16%) (*Gonçalves et al., 2021*) dynamometers are in the range of values observed for the custom rig (MDC = 11–31%), which suggests the custom rig has similar sensitivity to previously investigated devices. Nevertheless, the ranges of measurement error reported in the literature, including our custom rig, suggest that small-to-moderate improvements in hip muscle strength (2–40%) typically observed following 6-18 weeks of strength training (*Blazevich & Jenkins, 2002*; *Casartelli et al., 2018*; *Snyder et al., 2009*) or strength differences seen, for instance, between individuals with intra-articular hip pathology and controls (*Arokoski et al., 2002*; *Diamond et al., 2015*; *Kierkegaard et al., 2018*), may remain undetected. Thus, other measurements of hip muscle function (e.g., electromyography, musculoskeletal modelling) may be required in combination with hip muscle strength for comprehensive evaluation of hip muscle function.

Absolute torque measurements using the custom rig differed, on average, by 0–9% from the MDD for hip flexion, internal rotation, and external rotation. The large limits of agreement (77–131%) suggest measurements from both devices could elicit different results and should not be used interchangeably (*Bland & Altman, 2010*). Our observations are in agreement with previous studies that showed large disagreement between measurements from hand-held dynamometers and MDD (*Bazett-Jones & Squier, 2020*; *Martins et al., 2017*). However, strong relationships ($r = 0.86$ to $1.00$) have been found between load-cell and MDD force measures when both devices have simultaneously acquired data from the same trial (i.e., both the load cell and the MDD were attached to the participant) (*Romero-Franco, Jiménez-Reyes & Montaño Munuera, 2017*). Nevertheless, when the load cell measurement error (SEM) was compared between days, researchers showed similar results to those reported here (3.9 to 16.3%) (*Romero-Franco, Jiménez-Reyes & Montaño Munuera, 2017*). Together, these results suggest differences observed between devices in ours and other studies (*Bazett-Jones & Squier, 2020*; *Martins et al., 2017*) could be partly attributed physiological variation that is expected when measuring human subjects (e.g., fatigue, motivation, task familiarity) (*Caruso, Brown & Tufano, 2012*).

Measurement errors for the custom rig (MDC = 11–31%) were similar to the MDD (MDC = 9–20%), with the exception of hip extension (20% vs 9%). The MDD was attached at a short lever arm for hip extension whereas the load cell on the custom rig was attached at a long lever arm. The latter position could have allowed participants to recruit both the hip extensor and knee flexor muscles, as suggested by results of previous studies where a similar set-up was used for knee flexor strength testing (*Askling, Saartok & Thorstensson, 2006*; *Hickey et al., 2018*; *Yanagisawa & Fukutani, 2020*). Despite asking participants to avoid knee flexion and rigorously inspecting for knee flexion angle during hip extension strength assessments using the custom rig, participants may have used different strategies (i.e., hip or knee dominant) between devices, which could explain differences in measurement error. Thus, when using the custom rig, the ideal position to test hip extension strength still requires further investigation. Additionally, limits of agreement were generally larger than

the measurement error of either device, which suggests that differences between devices may be caused by true differences in task performance rather than measurement error. Nevertheless, the two devices appear to be measuring muscle strength originating from different movement strategies, thus comparisons between datasets from different devices should be made with caution.

We have uniquely demonstrated that unilateral maximal isometric hip strength can be assessed in intermediate planes with good reliability using a simple, inexpensive, and easily replicable experimental set-up. The load cell used (∼$400USD) is cheaper than a hand-held dynamometer (*Ishøi, Hölmich & Thorborg, 2019*; *Zhang et al., 2018*) or other commercially available devices (e.g., The GroinBar / ForceFrame) (*Desmyttere, Gaudet & Begon, 2019*; *O'Brien et al., 2019*) currently used to assess hip strength. However, the presented set-up was developed in a specialised biomechanics laboratory and some refinements would be required for it to be used in a clinical or athletic setting. A possible solution would be to fixate the load cell to an appropriately stable and supportive object (e.g., door or squat rack) as used previously (*Hickey et al., 2018*). Additionally, the recording devices used in our experiment can be substituted by an affordable and portable hardware and software (∼$100USD, e.g., USB-200 Series, MC measurement computing, Norton, MA, USA). This modified set-up would also allow for force to be acquired at other joints, though reliability would first need to be established.

This study has some limitations that warrant consideration. Only two testing sessions were performed, which may explain the moderate reliability of some strength measurements (*Hopkins, Schabort & Hawley, 2001*; *Koo & Li, 2016*). Further, only healthy young adults (23-33 years) were included, and it is unclear whether these results can be extrapolated to other populations with lower-limb musculoskeletal conditions. We also only tested one position for each strength task and, as previously suggested, body position can influence reliability measurements (*Bazett-Jones & Squier, 2020*; *Thorborg et al., 2010*) as well as muscle recruitment (*Glaviano & Bazett-Jones, 2020*; *Yanagisawa & Fukutani, 2020*). Thus, it remains unclear whether other positions could be more reliable or elicit higher levels of recruitment from the posterior/lateral hip muscles. Finally, we only compared the validity of the custom rig for three strength tasks (hip flexion, hip internal rotation, and hip external rotation) because these tasks had a similar set-up between devices. However, since these tasks showed poor agreement between devices, we anticipate the remaining tasks would show similar results.

## CONCLUSIONS

Good-to-excellent inter-session reliability was generally observed for maximal hip strength measurements performed in principal and intermediate planes using a custom rig. The measurement error associated with the custom rig was similar to that of an MDD, suggesting the custom rig may have utility in studies where large effect sizes are expected, though both devices may lack the sensitivity required to detect small changes in hip strength (<11–31%) commonly observed following intervention.

## ACKNOWLEDGEMENTS

The authors would like to thank all the volunteers who participated in the study and Alastair Quinn (Griffith University) for his contribution to the manuscript.

### Funding

The authors received no funding for this work.

### Competing Interests

The authors declare there are no competing interests.

### Author Contributions

- Basilio A.M. Goncalves and Laura E. Diamond conceived and designed the experiments, performed the experiments, analyzed the data, prepared figures and/or tables, authored or reviewed drafts of the paper, and approved the final draft.
- David J. Saxby and Rod S. Barrett conceived and designed the experiments, prepared figures and/or tables, authored or reviewed drafts of the paper, and approved the final draft.
- Adam Kositsky conceived and designed the experiments, performed the experiments, prepared figures and/or tables, authored or reviewed drafts of the paper, and approved the final draft.

### Human Ethics

The following information was supplied relating to ethical approvals (i.e., approving body and any reference numbers):

Griffith University's Human Research Ethics Committee granted Ethical approval to carry out the study within its facilities (GU Ref No: 2018/700).

### Data Availability

The data and code used to generate the results are available in the Supplemental Files.

### Supplemental Information

Supplemental information for this article can be found online at http://dx.doi.org/10.7717/peerj.11521#supplemental-information.

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
