# Peer review of "Reliability of hip muscle strength measured in principal and intermediate planes of movement"

_PeerJ, doi:10.7717/peerj.11521_

## Round 0.1 · original submission · Major Revisions

First of all, apologies for the delay in getting this paper reviewed and back to you. In the end we received three reviews of your paper and they were largely positive. While the decision is for major revisions, I believe the required changes are relatively easily achieved. I look forward to seeing the revised paper.

Reviewer 1 ·

Basic reporting

This article meets basic reporting standards, apart from an apparent error between the images and caption of figure 2, which is highlighted in the specific comments to the authors below.

Experimental design

The experimental design standards have been adequately met.

Validity of the findings

Findings appear to be valid and clearly reported to a high standard.

Additional comments

• Introduction
o Lines 39-40: please clarify the aspects of athletic performance you are referring to (e.g. sprint performance, jump height etc.)

• Methods
o Lines 87-88: please clarify if the exclusion criteria of having prior hip surgery is just within the last three months or lifetime history, as it currently isn’t absolutely clear
o Line 92: please clarify the rationale for allowing participants to complete their follow-up testing session within a range of 3 to 9 days after the initial session. There is potential that any muscle soreness resulting from the first session could still be present 3 days post the first, but this is unlikely at 9 days. Not a major criticism, but it needs to be addressed here in my opinion
o Line 94: should the cycle ergometer resistance be reported in Watts rather than Newtons?
o Line 96: please report the sample rate of the uniaxial load-cell in Hz
o Line 102: please clarify if the submaximal familiarisation contractions were standardised (e.g. 50% of maximal effort) or was no instruction given apart from being sub-maximal?
o Lines 112-114: please clarify why combined hip abduction and external rotation was not assessed? The response doesn’t need to be included in the paper, but I am curious as to why this combination was not included, given it is open prescribed as an exercise for people with hip pathology
o Lines 128-130: please clarify if and how the strap position was standardised relevant to the distal shank (e.g. proximity to the malleoli) and thigh (e.g. proximity to the femoral condyles) as this has important implications for the subsequent torque calculation
o Figure 2: the captions provided do not appear to match the lettering sequence of images in the figure. For example, the caption reads that A indicated hip abduction/adduction measures, but the images appear to be showing hip extension, which is references as C in the caption. Please correct this apparent error
o Figure 2: the image of hip extension using the custom rig appears to show the line of pull of the load cell and in-series elements to be very different to what would be expected to be generated by the movement of hip extension. Please clarify if this is the actual position of testing, as this would likely result in the knee flexors being a large contributor. If this image is not truly reflective of the testing setup, please correct and take another image

• Discussion
o Lines 274-277: consider citing Askling et al. who used a similar position to your hip extension custom rig setup but to measure knee flexion with an externally fixed load cell https://pubmed.ncbi.nlm.nih.gov/16371489/ This is an important point you correctly mention, as it is likely the custom rig setup was more an assessment of hamstring strength compared to short hip extensor muscle (e.g. glue max, adductor magnus) strength
o Line 289-290: consider citing Hickey et al. as an example of a study demonstrating the reliability of assessing knee flexor strength using load cells externally fixed to a squat rack https://pubmed.ncbi.nlm.nih.gov/29073840/

·

Basic reporting

no comment

Experimental design

no comment

Validity of the findings

no comment

Additional comments

Overall:
The current study assesses the reliability and validity of a custom dynamometer rig in customary planes and ‘intermediate’ planes. The manuscript is well written and provides thoughtful justification and discussion for exploring intermediate planes. I have a few suggestions for improvement listed below.
General comment:
The terms ‘planes’ and ‘directions’ are used interchangeably (e.g. line 54 & 73). Suggest chose one term (either planes or directions) and use throughout.
Specific comments:
Introduction
Line 54 – the second part of this sentence require references for these statements made: “tasks performed in intermediate plates generate the highest levels of activation” and “testing strength in directions outside of principal planes is necessary to fully…”
Line 59 – suggest replace the word ‘Normalising’ with another to avoid confusion since normalize is usually used in regards to adjusting force measures on a dynamometer to a participants height and or weight.
Line 68 – not sure what ‘this’ refers to – the antecedent that the pronoun refers to is not clear, please revise.
Line 69 – ‘combined actions’ and ‘function’ is slightly vague and can be specified by replacing with ‘force’ and ‘strength’. Please consider this or revising to clarify.
Line 71 – ‘other devices’ is vague – please replace with a specific set of technology, like HHD? Or remove.
Line 73 – Suggest instead of giving an example of an intermediate plane, define what you mean by intermediate plane… this term needs an operational definition somewhere in the introduction.
Line 76 – It is not clear how the term ‘absolute’ is different from maximum… please describe the meaning of absolute here, or revise.
Line 80-82 – The last sentence of the introduction is not related to the hypotheses, instead it is more of a discussion item. Suggest moving to the Discussion section.
Methods
Line 90 – Twenty participants meet the minimum sample size requirement for ICC as reported by Walter, et al. STATISTICS IN MEDICINE, VOL. 17, 101-110 (1998), mentioning this would increase the internal validity of the study.
Line 147 – Suggest moving the sentence that starts on this line to the procedures section
Line 155 – The word ‘relative’ seems unnecessary, suggest remove.
Line 165 – since torque is force multiplied by a constant (moment arm length), why analyze reliability with both? It seems redundant. Please add justification for this or revise to only assess force or torque.
Line 168 – equation may be missing a parenthesis between MDD and x100
Results
Line 191-194 – It is difficult to understand the results of Bland Altman plots clearly with variables in parentheses. Suggest explicitly state the results, rather than using parentheses to imply.
Discussion
Line 212-215 – Suggest revising this sentence to remove repetition of results.
Line 233 – A potential discussion point would be to compare the intra-session ICC reported by Aramaki et al., 2016 to the intra-session ICC of this study… since more than two repetitions were collected, an intra-session ICC can be calculated, and would provide a better comparison between the current study and previous literature.
Line 264 – This sentence mentions that multiple studies report large discrepancies between force measured using a HHD and MDD – however only one study was cited. Please provide additional citations or revise sentence to remove plural.

·

Basic reporting

Article is missing multiple references that would improve the quality of the manuscript.

Article does not provide adequate rationale for the need for the study, specifically the multiplanar measures.

At times the manuscript reaches in its conclusions.

Experimental design

Methods in some places could use more clear and thorough explanations.

Not clear about the knowledge gap that the research fills.

Not clear if it followed established reliability study guidelines.

Validity of the findings

Multiple comparisons not completed due to methods being different in one test versus another. Assume that the agreement would have been very poor with these results included.

Speculation is at times taken too far towards a clinical population.

Additional comments

The purpose of this study was to evaluate the inter-session reliability of hip strength tested in the cardinal and intermediate planes using a load cell and Biodex. The use of a load cell has not received extensive research in hip strength testing so this is an important topic due to certain limitations of HHD. The manuscript is generally well written. However, there are a number of concerns that the authors should address prior to the manuscript being considered for publication.

General Comments:

One of the biggest criticisms of this paper was that the need to test hip strength in these intermediate planes was not well argued. Lines 67-70 do not provide a convincing argument as to why strength needs to be measured in a multi-planar manner. Measuring strength in this fashion does not allow a clinician to target certain muscles and therefore, all hip muscles would need to be targeted. Testing multiple muscles takes more time and effort but provides more specific information. This needs to be better argued and explained.

Throughout the text, strength, force, and torque are used interchangeably. Force and torque should be used when discussing these measures and strength should be discussed when talking about the general idea or construct. Adjust accordingly throughout.

The discussion (and even the introduction) leans too heavily on the hip pathology literature even though this study is in a healthy population. Many references to hip strength testing are not cited with really only Thorborg articles being discussed. The following articles and subsequent discussion of them would help to improve this manuscript substantially.

References to consider:
ABD reliability & other considerations-
Krause et al., 2007
Kelln et al., 2008
Kawaguchi JB, 2010
Bazett-Jones et al., 2011
Krause et al., 2014
Romero-Franco et al., 2016 (this is using a load cell as well so important to compare)
Bazett-Jones et al., 2017
Martins et al., 2017 (already cited by not discussed)
Bazett-Jones & Squier, 2020

IR & ER reliability-
Katoh 2009
Katoh & Uchida 2011
Hoglund et al., 2014
Martins et al., 2017 (suggested above)
Bazett-Jones & Squier, 2020 (suggested above)
Glaviano & Bazett-Jones 2021 (EMG study)

Hip extension position selection-
Krause et al., 2014 (as above)
Kindel & Challis, 2017
Bazett-Jones & Squier, 2020 (suggested above)

Specific Comments:

L36 – the conditions that are listed here are hip, not LE. Hip strength testing also applies to PFP and other knee pathologies as well as ankle sprains and instability. Throughout the manuscript, the authors limit the impact of this study by only concentrating on the population of hip pathologies.

L68 – please clarify what “this motion” is referring to.

L75 – aims should use force or torque, not strength.

L77 – same as previous comment. Also need to specific which action since some were excluded from this analysis.

L84 – Were GRRAS guidelines (Kottner 2011) followed?

L99 – how was the dominant leg defined?

L111 – abbreviation of MDD not used here. Used consistently throughout or not at all.

L117 – the inconsistent positions are a major limitation of this study. For varying measures of hip ABD, 3 different positions are used when standing could have been used for all.

L128 & L131 - a more specifically defined placement should be provided since this can influence the measurements.

L133 – Was the joint angle confirmed to be the same as the angle given on the Biodex? The problem with this is that the pad that is attached causes the angle of the device to be different than the angle of the joint.

L138 - Was the weight of the test limb used to correct for gravity of the custom rig as well? How does this influence the data? Are you able to compare MDD and custom rig even if they both were not corrected?

L148 – Measurement of the lever arm should be a procedure, not data processing. Move up and describe more thoroughly.

L151 – was only the highest force value used or was the highest torque value used as well?

L153 – what were the results of the normality test?

L154 - Number of outliers excluded should be reported in results so we know the impact it has on the data.

L173 – Strength = force or torque?

L183 – good and excellent reliability should be reported separately as was above for the custom rig.

L185 – Weren’t the custom rig results from the principal planes provided in the first sentence? This would then be conflicting with that. Please clarify.

L191 – since there are only 3 comparisons, it seems that they all could be listed here and the bias direction could be named more explicitly (e.g. towards greater flexion measured by MDD)

L193 - Please provide 95% CIs for the LOA so that we know the precision of these.

L2019-211 – This statement seems like a reach since the authors did not test set-up reliability or inter-tester reliability and still had large measurement error. Not to mention that it requires a large, custom rig.

L212-218 – MANY other studies that have investigated hip strength reliability that could be added here. See general comment.

L227 – This study only measured healthy individuals, not those with hip pathologies so this statement seems like a reach.

L240 - This has been done somewhat similarly in Lee et al 2014 (https://pubmed.ncbi.nlm.nih.gov/24560168/) and McBeth et al 2012 (https://pubmed.ncbi.nlm.nih.gov/22488226/). This might improve the discussion.

L241 – application to hip pathology understanding is a stretch for this study. Suggest toning down language.

L254 – The Casarelli reference only applies to hip pathology. There are other pathologies that this statement may or may not apply to as well. This statement is too broad in its current form. Revise or remove.

L293 - The ability and desire of clinicians to figure out hardware and software would be questioned, especially with limited time.

L294 – the inability to compare half of the measures between the MDD and the custom rig leaves a large hole in the results of this study. This is a major limitation that needs to be addressed.

---

## Round 0.2 · Minor Revisions

Thank you for your revised paper. For the most part the reviewers are happy with the revisions. There are only a couple of minor changes now required. Please see the reviewer comments.

Reviewer 1 ·

Basic reporting

No comment.

Experimental design

No comment.

Validity of the findings

No comment.

Additional comments

Thank you to the authors for their considered responses to all of my previous comments. I am satisfied that they have adequately addressed all of my previous concerns, apart from one apparent referencing error in their in text response to my comment related to athletic performance. In their response to my comment they included an additional line addressing contribution of hip muscle strength to athletic performance and said they cited Prendergast et al., however, in the revised manuscript van Dyk et al. is cited, which does not appear to be the appropriate reference. As long as this minor error is corrected, I have no further comments and congratulate the authors on a thorough revised manuscript, which I believe warrants publication.

·

Basic reporting

no comment

Experimental design

no comment

Validity of the findings

no comment

Additional comments

Thank you for considering my revisions. The manuscript has greatly improved.

·

Basic reporting

No comment

Experimental design

No comment

Validity of the findings

No comment

Additional comments

The authors are commended on their much improved manuscript. All of my comments have been adequately addressed, except the introduction still seems to lack some justification. The "non-principle planes have never been measured" argument does not provide adequate justification, in my opinion . The third paragraph seems the least relevant (though I do appreciate the desire to have a clinical application). My suggestion is to write further about what Neumann 2010, Delp 1999, Dostal 1986, etc. have described as changing moment arms and/or length-tension, which can impact torques. When I think about the line of action for the posterior fibers of the glute med, as the hip is externally rotated (per your device and positions), the moment arm distance is increased and the muscle is shortened. How does the intermediate plane position impact the anatomical structures and mechanics of the hip? What is the rationale, from this specific perspective, for why testing in the intermediate planes would have any impact on the results? It seems like addressing this type of question that specifically provides a rationale for why differences might be found would be preferred over conjecture that these results are likely different from the primary planes just because the hip is complex and has 6DOF. However, if the editor feels that it is appropriate as is, I support his decision.

---

## Round 0.3 · accepted · Accept

Thank you for the quick turnaround on the last revisions.